# Magnetic properties of a capped kagome molecule with 60 quantum spins

**Roman Rausch[1][*], Matthias Peschke[2,3], Cassian Plorin[3,4] and Christoph Karrasch[1]**

**1** Technische Universität Braunschweig, Institut für Mathematische Physik,
Mendelssohnstraße 3, 38106 Braunschweig, Germany
**2** Institute for Theoretical Physics Amsterdam and Delta Institute for Theoretical Physics,
University of Amsterdam, Science Park 904, 1098 XH Amsterdam, The Netherlands
**3** I. Institute of Theoretical Physics, University of Hamburg, Notkestraße 9,
22607 Hamburg, Germany
**4** The Hamburg Centre for Ultrafast Imaging, Luruper Chaussee 149,
22761 Hamburg, Germany

* r.rausch@tu-braunschweig.de

## Abstract

We compute ground-state properties of the isotropic, antiferromagnetic Heisenberg model on the sodalite cage geometry. This is a 60-spin spherical molecule with 24 vertex-sharing tetrahedra which can be regarded as a molecular analogue of a capped kagome lattice and which has been synthesized with high-spin rare-earth atoms. Here, we focus on the $S = 1/2$ case where quantum effects are strongest. We employ the SU(2)-symmetric density-matrix renormalization group (DMRG). We find a threefold degenerate ground state that breaks the spatial symmetry and that splits up the molecule into three large parts which are almost decoupled from each other. This stands in sharp contrast to the behaviour of most known spherical molecules. On a methodological level, the disconnection leads to "glassy dynamics" within the DMRG that cannot be targeted via standard techniques. In the presence of finite magnetic fields, we find broad magnetization plateaus at 4/5, 3/5, and 1/5 of the saturation, which one can understand in terms of localized magnons, singlets, and doublets which are again nearly decoupled from each other. At the saturation field, the zero-point entropy is $S = \ln(182) \approx 5.2$ in units of the Boltzmann constant.



# 1   Introduction

Interacting quantum spins have a tendency to form singlet states, which have no preferred direction and minimize the antiferromagnetic exchange energy. This is captured by the Heisenberg Hamiltonian

$$H = \sum_{i<j} J_{ij} \mathbf{S}_i \cdot \mathbf{S}_j\,, \tag{1}$$

where $J_{ij}$ are the exchange couplings among $L$ spins, and $\mathbf{S}_i = \left(S_i^x, S_i^y, S_i^z\right)$ is a vector of spin-$S$ operators. This singlet formation is frustrated on non-bipartite lattices, among which vertex-sharing triangular geometries (kagome-type) and vertex-sharing tetrahedral geometries (pyrochlore-type) stand out as particularly complicated and interesting. Such systems can be roughly grouped into (i) 1D chains, (ii) 2D/3D lattices, and (iii) finite molecules. Among the molecules, ferric wheels are analogous to 1D chains or ladders [1,2], while hollow cages [3–10] (such as the Platonic or Archimedean solids) are analogous to 2D planes, albeit with a spherical topology.

In this work, we focus on the physics of quantum spins in molecular systems. One of the most well-studied molecules is the icosidodecahedron, a molecular analogue of the kagome lattice [3,5–9]. This 30-site spherical cage can be formed by transition metal ions $V^{4+}$, $Cr^{3+}$, $Fe^{3+}$ in the Keplerate molecules [11–13] with $S = 1/2, 3/2$, and $5/2$, respectively. Recently, a cage-like molecule with $L = 60$ spins was synthesized that is based on vertex-sharing tetrahedra [14] and that can be classified as a molecular analogue of a capped kagome compound [15–17] (see Fig. 1). The addition of the "caps" promotes the triangles to tetrahedra and is a step towards the 3D pyrochlore lattice.

Due to the high frustration and three-dimensionality of the pyrochlore lattice, not much is known about the ground state of the isotropic Heisenberg model on this geometry. Neither the value of the ground-state energy nor the existence of a spin gap have been reliably estimated [18,19] despite a wealth of approaches. By using extrapolation schemes from low to high temperatures, a gapless spectrum and a value for the energy has been proposed recently [20]. Exact diagonalization reaches its limits with about 36 sites [21,22] and finds a disordered ground state. On the other hand, approximate results (often based on weakening the intertetrahedra coupling $J'$ to obtain a small expansion parameter) indicate lattice symmetry breaking [23–27]. However, such methods may not properly take into account the competition between different phases. Recent progress involves the application of the pseudofermion functional renormalization group [18], where such competition is thought to be treated more

faithfully and which again points to a disordered ground state. An approach coming from the high-temperature region comparing various imaginary-time propagation techniques [19] indicates that much of the entropy is unreleased before low temperatures can be reached, pointing towards a high density of states close to $T = 0$. One should note that in contrast to the 3D pyrochlore lattice, a 60-spin molecule can be treated accurately using the density-matrix renormalization group (DMRG), while still having a nontrivially large size.

In experimental realizations of the capped kagome molecule [14], the spin centres are Gd atoms with $S = 7/2$ (Dy, Er and Y were also used [28,29]). This allows for an approximation with classical spins, and it was shown that the system can be described well by the classical isotropic Heisenberg model [14]. While the absence of a strong anisotropy prevents Ising-like ordering and is a prerequisite to observe quantum effects, such effects are washed out by the large value of $S$. This motivates us to look at the same geometry for the case of $S = 1/2$, where quantum fluctuations are the strongest.

There are several scenarios for the nature of the ground state of such a frustrated spin system. One possibility is an ordered state which breaks the spin symmetry and which is found, e.g., for the triangular lattice [30–33]. Another possibility is a "valence-bond solid" (VBS) in which translational invariance is broken by a particular pair-singlet covering. However, spin symmetry remains unbroken, so that the total spin $S_{\text{tot}}$ obtained from

$$\left\langle \mathbf{S}_{\text{tot}}^2 \right\rangle = \sum_{ij} \left\langle \mathbf{S}_i \cdot \mathbf{S}_j \right\rangle = S_{\text{tot}} (S_{\text{tot}} + 1) \tag{2}$$

is zero. A VBS state tends to appear for fine-tuned parameters or very small systems [34–37], though there are notable exceptions [38]. Yet another possibility is that the ground state is highly degenerate due to the exponentially large number of combinations to distribute pair-singlets in 2D and 3D [39]. However, this degeneracy tends to split into a unique "liquid-like" ground state with exponentially decreasing correlations and many low-lying singlet states. The latter case is what is found for frustrated polyhedra, such as the icosahedron ($L = 12$) [4], the cuboctahedron ($L = 12$) [5,6], the dodecahedron ($L = 20$) [4], and the icosidodecahedron ($L = 30$) [3,5–9]. They have nondegenerate ground states that transform according to the trivial irreducible representation $A_{1g}$ of the icosahedral group $I_h$ or the octahedral group $O_h$; as well as a number of low-lying $S_{\text{tot}} = 0$ states that grows quickly with the size.

In this paper, we will show that unlike these smaller polyhedra, the ground state of our large capped-kagome molecule is not given by the trivial irreducible representation class $A$, but rather by $T$. Because $T$ is three-dimensional, this makes the ground state threefold degenerate, which goes hand in hand with a spatial symmetry breaking, as we argue below. Three orthonormal basis states can be conceptualized as follows: The two poles and a belt around the equator of the sphere nearly completely decouple from each other and the rotational symmetry is reduced to rotations about only one coordinate axis. The different ground states are thus related by a global reshuffling of the spins of the whole molecule which cannot be achieved with local operations in reasonable time and which leads to a "glassy" behaviour for the DMRG algorithm (which hinges on local updates). To the best of our knowledge, such a state has not been found elsewhere and is thus a new addition to the list of possible scenarios for the ground states of frustrated geometries.

After computing the ground state, we analyze the behaviour of several physical quantities. We demonstrate the existence of localized magnons, resulting in a zero-point entropy of $S = \ln(182) k_B \approx 5.2 k_B$ per molecule ($k_B$: Boltzmann constant) at the saturation magnetization. We observe wide magnetization plateaus at 3/5 and 1/5 of the saturation, which can be explained by commensurate numbers of spinflips that can form localized confined singlet or doublet states. This can be seen as a generalization of localized magnons.

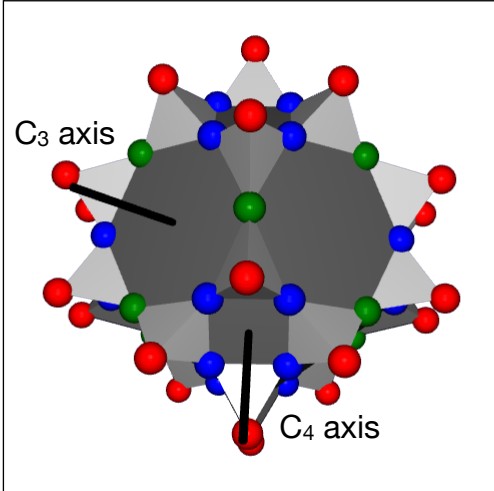
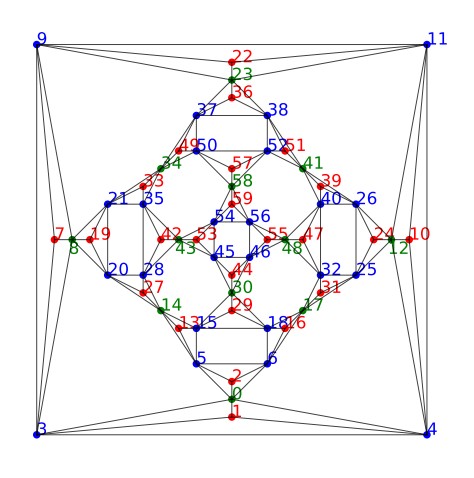

Figure 1: Left: Ball-and-stick drawing of the SOD60 molecule with a 4-fold and a 3-fold symmetry axis indicated. Right: Projection on the plane (Schlegel diagram) using the square orientation. The enumeration of the sites is the result of applying the Cuthill-McKee compression. Equivalent sites are drawn in the same colour.

## 2 Geometry

In a recent work, various hollow cages with magnetic centres have been synthesized, the largest of which has $L = 60$ spin sites [14]. This cage can be understood by starting with a *rectified truncated octahedron* [40]. The truncated octahedron is a well-known Archimedean solid, while the *rectification* procedure is a "shaving off" of the vertices of a polytope, such that the stubs share a vertex. In this case, it results in 8 hexagon faces, 6 square faces and 24 vertex-sharing triangle faces. Furthermore, each of the 24 triangles is "capped" (or "stellated") with an additional spin site, forming vertex-sharing tetrahedra. Thus there are 36 "base spins" residing on the vertices of the polytope and 24 "apex spins" on top of the triangles. These two layers can also be thought of as a kagome-lattice layer and a triangular-lattice layer. In a different chemical context, this object is known as a "sodalite cage" [29,41], commonly abbreviated as SOD. We thus use the shorthand "SOD60" to refer to this molecule. The geometry is depicted in Fig. 1.

There are three inequivalent sites which we depict as red, green, and blue balls in Fig. 1: (r) the apices of the tetrahedra, (g) the vertices bounded by two hexagons and two base triangles, (b) the vertices bounded by a hexagon, a square and two base triangles.

One finds that there are four inequivalent nearest-neighbour bonds, corresponding to the connections (r)-(g), (r)-(b), (g)-(b) and (b)-(b). We note that the triangles are isosceles, with the long edges exceeding the short ones by a factor of $\sqrt{6}/2 \approx 1.22$. One can therefore expect that this leads to slightly different exchange constants $J$, but as a first approach, we assume a homogenenous value of $J \equiv 1$ for all nearest neighbours of the interaction graph $J_{ij}$. The symmetry group of the molecule is $O_h$ (octahedral) and has the irreducible representation classes $A$ (1), $E$ (2), $T$ (3), where the brackets indicate the multiplicity. More precisely, the irreducible representations are: $A_{1g}, A_{1u}, A_{2g}, A_{2u}, E_g, E_u, T_{1g}, T_{1u}, T_{2g}, T_{2u}$. There are three $C_4$ symmetry axes (as in a cube), as well as four $C_3$ symmetry axes (see Fig. 1).

The maximal distance of the spin-spin correlations is $d = 7$ and there are 144 nearest-neighbour bonds.

We also introduce a new hypothetical cage "SOD20"[1], where the capping procedure is extended to the triangles of the cuboctahedron, resulting in 12 base spins and 8 apex spins (see Fig. 4). This leads to a system with $L = 20$ spins, which can be readily solved in the full Hilbert space by the Lanczos algorithm, while having a similar geometry and also belonging to $O_h$. This is useful as a small system that one can compare to SOD60. We are not aware of the existence of such a structure, but a cuboctahedron where the squares are capped instead of the triangles does exist as a Fe-based magnetic molecule [42, 43].

## 3  Technical details

In order to find the ground-state wavefunction of the Hamiltonian (1) with $J_{ij} \equiv 1$ for the bonds depicted in Fig. 1, we employ the DMRG algorithm, which provides a highly accurate way to variationally determine the ground state within the class of matrix-product states [44]. The dimension of the matrices – the so-called bond dimension – is a measure of the entanglement and serves as the key numerical control parameter. The reason why DMRG can tackle exponentially-large Hilbert spaces is that many ground states are only entangled locally ("area law") and can thus be represented faithfully by matrix-product states with a small bond dimension. Our code fully exploits the SU(2) spin symmetry [45] of the problem. The maximal SU(2)-invariant bond dimension is $\chi_{\mathrm{SU(2)}} = 7000$, which corresponds to an effective bond dimension of about $\chi \sim 30000 - 34000$ when SU(2) is not exploited. Convergence of the algorithm is assessed by computing the energy variance per site

$$\Delta E^2/L = \left(\left\langle H^2\right\rangle - \left\langle H\right\rangle^2\right)/L \,. \tag{3}$$

The interaction graph given by $J_{ij}$ is compressed by applying the Cuthill-McKee algorithm [46], which reduces the graph bandwidth to 16. In physical terms, this corresponds to the maximal hopping distance on the effective 1D chain geometry that is required by DMRG. The resulting numbering of the sites is displayed in Fig. 1. We refer to Ref. 10 for a discussion of the dependence of the results on the numbering. We find that the matrix-product-operator (MPO) representation of the Hamiltonian can be compressed without losses [47] down to a maximum size of $23 \times 20$.

## 4  Degenerate ground state

The left part of Fig. 2 shows the nearest-neighbour spin-spin correlations in the ground state obtained by DMRG. Evidently, the ground state is symmetry-broken: Instead of the three $C_4$ rotational symmetry axes that pierce the square faces, we are only left with one, while the others are reduced to $C_2$ axes. The $C_3$ symmetries along the axes that pierce the hexagons are all completely broken. This suggests a threefold degeneracy according to the irreducible representation $T$. We thus expect two other ground states to exist that have similarly broken symmetries along the other two coordinate axes.

After computing one member $\left|E_0\right\rangle$ of the ground-state manifold, the full multiplet can be obtained within the DMRG by setting

$$H' = H + E_p\left|E_0\right\rangle\left\langle E_0\right|, \tag{4}$$

where $E_p$ is a sufficiently high energy penalty. The ground state of $H'$ is then a different member of the multiplet (or the first excited state in case of a nondegenerate ground state). We find, however, that this technique fails in our case even though we perform two-site sweeps

---

[1]We note that SOD20 is distinct from the $Gd_{20}$ system of Ref. 14, which is just a dodecahedron.

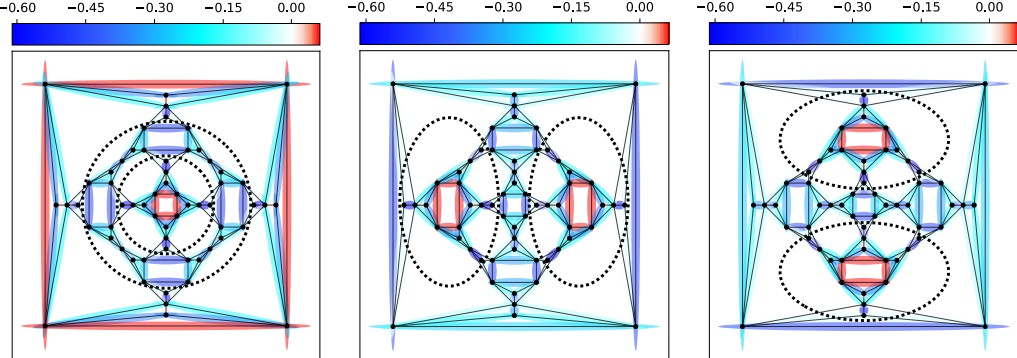

Figure 2: Nearest-neighbour spin-spin correlations $\langle \mathbf{S}_i \cdot \mathbf{S}_j \rangle$ for the three symmetry-broken ground states. The dotted lines indicate where parts of the molecule nearly decouple.

and apply standard methods of adding fluctuations [44]. The algorithm always converges to one of many low-lying singlet states whose energy is larger than $E_0$. We will investigate the physical reason for this failure in the next section.

To obtain the full multiplet, we need to proceed in a different way. We explicitly perform a spatial rotation of the state $\left| E_0 \right\rangle$ such that one ends up with a state that should correspond to one of the other two members of the ground-state manifold. On a technical level, this can be achieved by a sequence of transpositions (see App. A for details). For $S = 1/2$, each transposition is carried out by applying the permutation operator [48]

$$P_{12} = 2\mathbf{S}_1 \cdot \mathbf{S}_2 + \frac{1}{2}. \tag{5}$$

Acting with $P_{12}$ on an antisymmetric pair-singlet (symmetric pair-triplet) state gives $-1$ $(+1)$ as an eigenvalue. We find that 45 transpositions are necessary for a rotation by 90 degrees. Such a large product of operators cannot be easily handled in an MPO representation. The bond dimension increases after each transposition, which makes truncations necessary and introduces errors. The energy of the rotated state thus becomes significantly higher than that of the ground state. However, the result can be used as a starting guess for another DMRG ground-state calculation governed by $H$, which allows us to determine the ground-state manifold $\left| E_0^{(a)} \right\rangle$, $a = 0, 1, 2$, to a satisfactory accuracy. The three ground states are orthogonal to about $\left\langle E_0^{(a)} \middle| E_0^{(b)} \right\rangle = \mathcal{O}\left( 10^{-5} \right)$ $(a \neq b)$, and the energy per spin agrees within four digits (see Tab. 1). The resulting spin-spin correlations are presented in the central and right part of Fig. 2, where the other two expected symmetry axes are now apparent. Averaging over the spin-spin correlations

$$\overline{\langle \mathbf{S}_i \cdot \mathbf{S}_j \rangle} = \frac{1}{3} \sum_{a=0}^{2} \left\langle E_0^{(a)} \middle| \mathbf{S}_i \cdot \mathbf{S}_j \middle| E_0^{(a)} \right\rangle, \tag{6}$$

we find that the spatial symmetries are restored, which is shown in Fig. 3. In total, this provides conclusive evidence for the existence of a degenerate, symmetry-broken ground state[2]. We stress that this is not an artifact of the numerical method: Once the state is well-approximated by a matrix-product state (which is ensured by a small energy variance), the breaking of the

---

[2]In principle, one can determine which irreducible representation ($T_{1g}$, $T_{2g}$, $T_{1u}$, or $T_{2u}$) is associated with the ground-state manifold by computing the corresponding characters. This requires the evaluation of expectation values $\left\langle E_0^{(a)} \middle| C \middle| E_0^{(a)} \right\rangle$, where $C$ represents a particular rotation or spatial inversion. Since $C$ is either a very large MPO or a product of many MPOs, we find that such a calculation is not feasible due to the prohibitively large bond dimension.

Table 1:   Total energy and energy per spin of the three symmetry-broken ground states, from which $E_0/L = -0.431(7)$ can be estimated.  The last column shows the energy variance per site, Eq. (3).

| $a$ | $E$ | $E/L$ | $\Delta E^2/L$ |
|---|---|---|---|
| 0 | -25.900473 | -0.43167 | $5.6 \cdot 10^{-5}$ |
| 1 | -25.895744 | -0.43160 | $3.6 \cdot 10^{-4}$ |
| 2 | -25.897953 | -0.43163 | $2.1 \cdot 10^{-4}$ |

Table 2:  Average of the spin-spin correlations for the inequivalent bonds via Eq. (6). The errors are given by the standard deviation of the distribution over the bonds, and the colour labels correspond to the coloured sites in Fig. 1.

| Bond $b$ | $\overline{\langle \mathbf{S} \cdot \mathbf{S} \rangle}_b$ |
|---|---|
| red-green | $-0.3241 \pm 0.0094$ |
| red-blue | $-0.1804 \pm 0.0060$ |
| green-blue | $-0.0798 \pm 0.0029$ |
| blue-blue | $-0.2345 \pm 0.0073$ |

spatial symmetry seen in Fig. 2 is the smoking-gun evidence for a ground-state degeneracy, and constructing the full multiplet serves as an additional corroboration.

We remark that symmetry breaking has to be taken with the usual caveat for finite systems: For finite temperatures, the free energy of a symmetry-broken system has degenerate minima with energy barriers between them. If the system is initially confined to one minimum, it has some probability to tunnel to another one, as long as the barrier remains finite, so that the symmetry breaking is not persistent. In the thermodynamic limit, the barrier becomes infinite and the system is perfectly dynamically isolated. For a finite system, this dynamical isolation is only approximate, but the isolation time should become large for large systems (as we have here), as well as for sufficiently small temperatures.

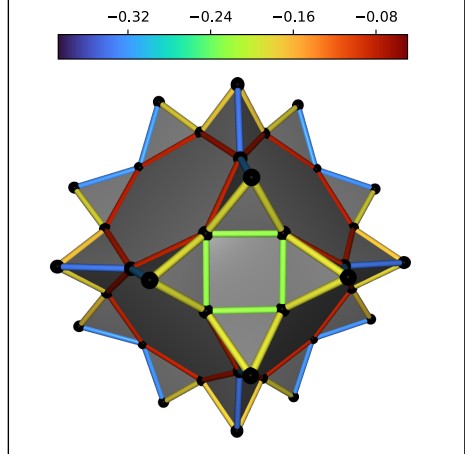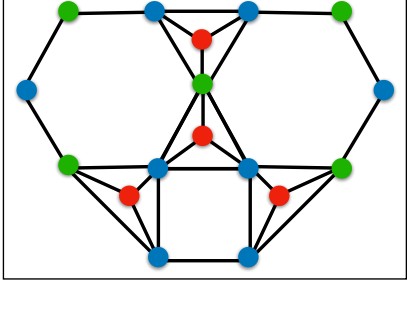

Figure 3:  Left: An average of the nearest-neighbour spin-spin correlations across the three ground states via Eq. (6) restores the spatial symmetry. Right: Neighbourhood of a tetrahedron for reference. The colour conventions are as in Fig. 1.

## 5 Nearly disconnected subsystems

The physical reason behind the failing of the projection technique in Eq. (4) becomes apparent when examining the spin-spin correlations in Fig. 2 more closely. The dotted lines intersect the bonds where the correlations are very small, around $-0.027$ for the red-blue bonds and $-0.0076$ for the blue-green bonds. From this one can see that the molecule breaks up into three nearly decoupled parts, 16 spins on the north and south pole, respectively, as well as 28 spins on a belt along the equator.

There are some ways to further characterize this behaviour quantitatively: For example, calculating the total spin of the decoupled parts, we find $\left\langle \mathbf{S}_{\mathrm{tot}}^2 \right\rangle \approx 0.15$ for the 16-spin clusters and $\left\langle \mathbf{S}_{\mathrm{tot}}^2 \right\rangle \approx 0.3$ on the 28-spin cluster, indicating that these subsystems are themselves almost singlet states. Furthermore, by computing the ground-state energies of the two poles and the equator separately, we find $[2E_0(\text{pole}) + E_0(\text{equator})]/L = -0.4294$, or about 99.5% of the exact energy density.

This phenomenology is reminiscent of a VBS state. However, the decoupled parts are not just pairs of sites, but large subsystems which are positioned at different locations for each member of the ground-state manifold. Hence, two different members of the ground-state manifold can only be connected by a global rearrangement of basically all the spins of the system. It now stands to reason that this is difficult to achieve with local DMRG updates. Instead, the approach yields local excitations of the disconnected parts. This is similar to what is usually called "glassy" behaviour: While a state of lower energy exists, the algorithm is frozen and has trouble finding it with only local updates and with local interactions. Such behaviour also underlies the anisotropic ferromagnetic Ising model on the pyrochlore lattice (commonly known as "spin ice"): Theory predicts an extensive ground-state degeneracy due to the strong frustration, which contradicts the third law of thermodynamics. One thus expects that a small perturbation will break the degeneracy and prefer a certain configuration, yet the degeneracy is also measured experimentally. The reason seems to be that approaching the true ground state requires a large number of spin flips, which is improbable and does not happen on the experimental timescale [49]. This leaves the system trapped in various local minima, similar to how the DMRG algorithm is trapped when trying to solve Eq. (4).

We might in fact also compare the situation with intrinsic topological order, which is found for the toric code model or for quantum dimer models in 2D [49–52]. In such a state, the ground-state degeneracy depends on the topology of the space the system is confined to, and each member of the ground-state manifold has a distinct winding number. This winding number is preserved exactly and cannot be changed by the Hamiltonian. In our case, the disconnection is only approximate, i.e., connecting the ground states is difficult in practice by a local Hamiltonian and only with local updates.

We point out that a symmetry-broken ground state with nearly disconnected parts only appears for a system that is large enough and thus constitutes a many-body effect. Figure 4 shows the nearest-neighbour spin-spin correlations of the smaller SOD20 molecule, which can be solved using exact diagonalization. We find a unique ground state with $E_0/L = -0.43440$ with no broken symmetries.

Finally, we remark that exactly confined states are also known from the solution of the tight-binding Hamiltonian on the Penrose lattice [53, 54], which is, however, bipartite.

### 5.1 Nearest-neighbour valence bond picture

One attempt to make sense of interacting quantum spins is the nearest-neighbour valence bond picture [55] (NNVB), where one restricts the Hilbert space to singlet pairs between nearest neighbours and seeks the solution as a superposition of these. In particular, a resonance

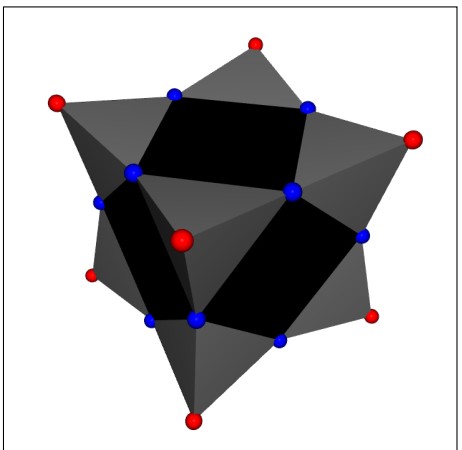 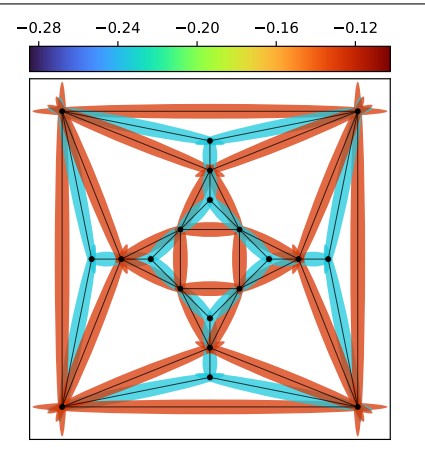

Figure 4: Left: Ball-and-stick drawing of the hypothetical SOD20 molecule (a cuboc-tahedron where each triangle face is decorated (capped) with an additional apex spin site). The sites that are distinct by symmetry are coloured red (apex) and blue (base). Right: Nearest-neighbour spin-spin correlations on this geometry. The two distinct values that appear are: $-0.2346$ (red-blue) and $-0.1274$ (blue-blue). The ground state is unique with no broken symmetries. The results were obtained using exact diagonalization.

between parallel bonds can be especially effective in reducing the energy [55].

In the case of SOD60, parallel bonds are found on the square plaquettes (blue-blue) and this may explain their relatively large correlations (see Tab. 2) at the expense of the red-blue and green-blue ones. This leaves the red (apex) spins to couple more strongly with the green spins. On the other hand, we note that for SOD20 (Fig. 4), the square plaquettes show weak correlations.

We may also attempt to understand the VBC-like patterns: The number of all NNVB states is given by the Hafnian of the interaction matrix $J_{ij}$ [56]. For SOD60, using [57] we obtain $\mathrm{haf}[\underline{J}] = 5,971,817$ and for the subsystems $\mathrm{haf}[\underline{J}(\mathrm{pole})] = 2$, $\mathrm{haf}(\underline{J}[\mathrm{equator}]) = 800$. We conclude that there are only $2 \cdot 2 \cdot 800 = 3600$ NNVB configurations that do not cross the boundaries (or about 0.06%). Thus, the reason for the disconnection patterns does not seem to relate to the paucity of NNV bonds that cross the subsystem borders.

We remark that for tetrahedra-based lattices, linear independence of NNVB states does not hold [58], since it already breaks down locally for a single tetrahedron. Thus, the NNVB picture seems only of limited use in this case.

# 6   Finite magnetic fields

We now study the properties of SOD60 in the presence of a finite magnetic field $B$. In Fig. 6, we show the magnetization $M = S_{\mathrm{tot}}$ as a function of $B$ in the ground state of SOD60 as well as of the hypothetical SOD20 molecule. The results were obtained by computing the lowest energy state in each sector of the total spin $S_{\mathrm{tot}}$ with an SU(2)-invariant bond dimension of $\chi_{\mathrm{SU}(2)} = 3000$ (which, e.g., corresponds to $\chi \sim 85000$ in the sector with $S_{\mathrm{tot}} = 18$ if no symmetries are exploited).

We observe wide magnetization plateaus that appear at 1/5, 3/5, and 4/5 of the saturation value. Their broadness implies that they are thermodynamically stable and should be observ-

able in the experiment. Such a signature could serve as a check that a given system can indeed be described by an isotropic $S = 1/2$ Heisenberg model. We note that a wide 3/5 plateau was experimentally observed in a capped kagome chain with $S = 1/2$ based on Cu [17], though its ground state was found to have long-ranged canted antiferromagnetic order.[3]

We will now try to understand the reason for the appearance of the wide magnetization plateaus as well as the nature of the corresponding fractions. At large fields, this can be achieved by using the picture of localized magnons.

## 6.1 Localized magnons

The emergence of localized magnons due to frustration is an effect that is described in detail in various publications [60–65]. Here, we focus on the essential quantitative properties for the SOD60 molecule. In short, an eigenstate of the system one spinflip away from the saturation ($S_{\text{tot}} = L/2 - 1 = 29$, $M = S_{\text{tot}}$) can be analytically expressed as:

$$\left|\Psi_{\text{LD}}\right\rangle = \sum_{l(i) \in \text{LD}} (-1)^{l(i)} S^{-}_{l(i)} \left|\uparrow\uparrow\dots\uparrow\right\rangle, \tag{7}$$

where $S^{-}_i = S^x_i - i S^y_i$ is the spinflip-down operator and LD denotes the bipartite "localization domain" of the magnon. In our case, the LD is a circular unfrustrated path of sites, consecutively numbered $l = 0, 1, 2, \dots$, which is sketched in Fig. 5. The proof that the above expression is an eigenstate is a matter of standard quantum mechanics. Proving that it is also the lowest-energy state in the sector with $S_{\text{tot}} = L/2 - 1$ is more difficult [60], but can be readily verified numerically. The localization effect can be understood in terms of destructive interference: The spinflip terms that would otherwise let the magnon propagate through the entire lattice cancel exactly if the localization domain is bounded by triangles. The magnon is thus forced to "run in a circle" on the LD sites with a momentum of $k = \pi$.

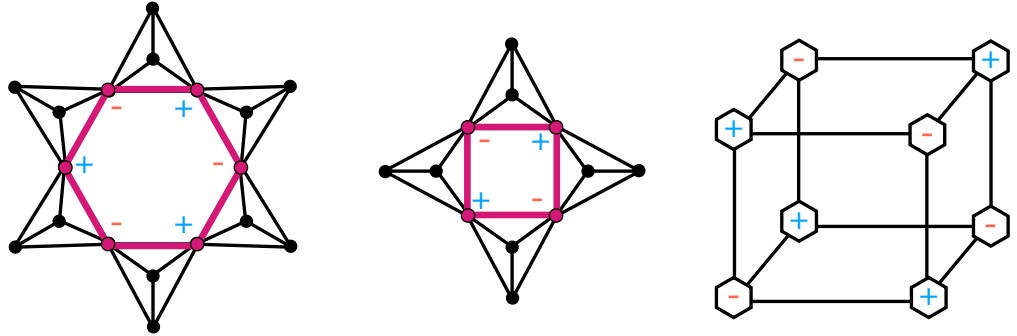

Figure 5: The possible magnon localization domains (LDs) of the SOD60 molecule on the hexagons and squares (see Sec. 6.1). The $\pm$ sign indicates the amplitude in Eq. (7).

The right side is an abstracted way to understand the distribution of the LDs on the molecule: The 8 hexagon plaquettes form the corners of a cube, while the 6 square plaquettes are identical to the square faces of the cube. The $\pm$ sign refers to the superposition of localized magnons in Eq. (8).

---

[3]Theoretically, one expects a width of $0.75 - 7.5$ T if one assumes that $J$ is in the range $J/k_B \sim 1 - 10$ K [59] and that the gyromagnetic ratio is $g = 2$. For Gd-based SOD60, however, a very weak $J/k_B \approx 0.15$ K was estimated [14], which is typical of rare earths and translates into a plateau width of 0.1 T. We note that in the experiments of Ref. [17], the 3/5 plateau of the Cu-based compound seems to span at least 8 T.

For SOD60, we have 14 localization domains given by the 6 squares and the 8 hexagon faces (see Fig. 5). The change in energy from the fully polarized state (with $E = 144/4 = 36$) due to the presence of one magnon is $\Delta E = 4$. We can continue to add up to $N_\downarrow = 6$ magnons that remain noninteracting on spatially separated squares and hexagons. The ground-state energy for fixed $S_{\text{tot}} = L/2 - n$, $n = 0, 1, \ldots 6$, is thus of the linear form $E = (36 - 4n)$. The corresponding ground-state degeneracies are presented in Tab. 3. They are related to the number of linearly independent ways to arrange the magnons on the localization domains of the system. The values are thus not obvious, but can be determined using exact diagonalization. We have also confirmed them using DMRG, which additionally validates our code.

In the regime $S_{\text{tot}} = L/2 - n$, $n = 0, 1, \ldots, 6$ the ground-state energy in the presence of a magnetic field, $E_M(B) = 36 - 4(30 - M) - B \cdot M$, forms a family of curves for different magnetizations $M \equiv S_{\text{tot}}$ that all intersect at the saturation field of $B_{\text{sat}} = 4$. Above (slightly below) the saturation field, the fully polarized state with $M = L/2 = 30$ (the state with $M = L/2 - 6 = 24$) is the ground state. The states with values of $M$ in between are never the ground state. We thus have a magnetization jump from $M = M_{\text{sat}} = 30$ to $M = 24 = 4/5 \cdot M_{\text{sat}}$. This is can be seen in Fig. 6.

At $B_{\text{sat}} = 4$, all the subspaces become degenerate, and the total degeneracy of the ground state is given by the sum of all magnon subspaces, $N_{\text{deg}} = 182$. Hence we obtain a zero-point entropy of $S = \ln(182) k_B \approx 5.2 k_B$ per molecule (or $0.087 k_B$ per spin). For comparison, on the icosidodecahedron, $S = \ln(38) k_B \approx 3.64 k_B$ per molecule (or $0.121 k_B$ per spin) can be achieved. When the field is varied close to the saturation, the large change in entropy results in an enhanced magnetocaloric effect [64].

The fact there are only 13 instead of 14 localized magnons in the $S_{\text{tot}} = 29$ subspace can be seen as follows: Ignoring the apex spins, the molecule can be thought of as a cube with the hexagon plaquettes being placed at the corners and the square plaquettes being placed at the faces (see Fig. 5). Since the hexagons form a bipartite lattice, we can enumerate them with even and odd numbers for the respective sublattices. Then the following relation holds:

$$\sum_i (-1)^i \left| \Psi_{\text{hexagon},i} \right\rangle \propto \sum_j \left| \Psi_{\text{square},j} \right\rangle. \tag{8}$$

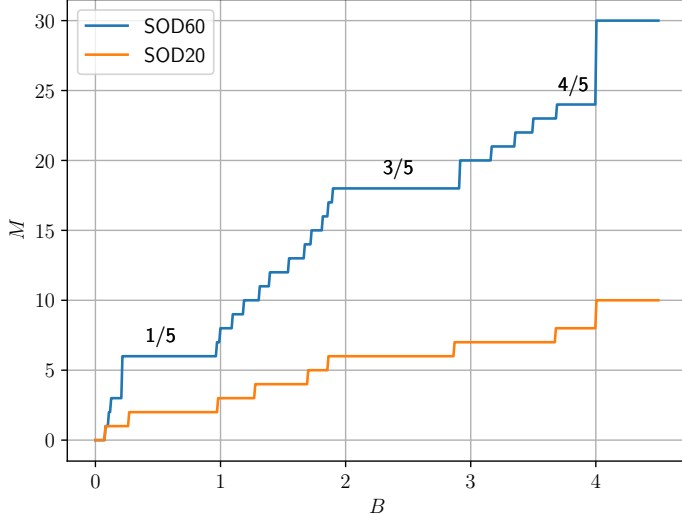

Figure 6: Magnetization $M = \sum_i \left\langle S_i^z \right\rangle$ as a function of the applied magnetic field $B$ in the ground state of the SOD60 as well as of the SOD20 molecule.

Table 3: Values of the lowest energy for total-spin values close to full saturation ($S_{\text{tot}} = 30$), as well as the corresponding degeneracies. Using SU(2) symmetries, we have set the $S_{\text{tot}}$ quantum number rather than explicitly ramping up a magnetic field. For $S_{\text{tot}} = 29$, there are 13 linearly independent ways to place one localized magnon on the 6 squares and 8 hexagons (see Eq. 5). For each downstep of $S_{\text{tot}}$, the number of magnons increases by one, the energy decreases linearly, while the number of combinations grows rapidly and peaks at "half-filling" or 3 magnons. For $S_{\text{tot}} = 24$, there is just one combination of arranging the 6 magnons by placing them on all the squares. The effect stops at that point, as can be seen from the deviation from the linear behaviour of the energy at $S_{\text{tot}} = 23$.

| $S_{\text{tot}}$ | $E_0(S_{\text{tot}})$ | $N_{\text{deg}}$ | $N_{\text{magnon}}$ |
|---|---|---|---|
| 30 | 36 | 1 | 0 |
| 29 | 32 | 13 | 1 |
| 28 | 28 | 55 | 2 |
| 27 | 24 | 71 | 3 |
| 26 | 20 | 25 | 4 |
| 25 | 16 | 16 | 5 |
| 24 | 12 | 1 | 6 |
| 23 | 8.31(6) | 1 | - |

Since the hexagons share one site, their amplitudes are cancelled by the factor of $(-1)^i$, so that the staggered superposition of the hexagon-magnons becomes proportional to the superposition of the square-magnons, revealing the linear dependence.

## 6.2 Localized singlets and doublets

The plateaus at $M/M_{\text{sat}} = 3/5$ and $M/M_{\text{sat}} = 1/5$ can be thought of as an extension of the previous concept from localized magnons to localized singlet clusters: The fraction of 3/5 corresponds to $N_\downarrow = 12$ spinflips, which can be arranged in an antiferromagnetic fashion on the square faces. Instead of localized one-magnon states, we now have clusters with $\langle \mathbf{S}_i \rangle \approx 0$ (see Fig. 7). They form a commensurate distribution on the molecule geometry and optimize the antiferromagnetic exchange energy, thus effectively resisting a change in magnetization when a field is applied.

We note that such states were also observed in the octahedral Heisenberg chain, where the localization domains are squares as well [66–70]. The concept of localized magnons can be extended to these two-magnon states at low fields, which allows for a classical dimer approximation to treat the thermodynamics [67, 68, 70].

In contrast, $N_\downarrow = 18$ spinflips (2/5 configuration) do not lead to an optimal arrangement and do not produce a plateau. For the next special value of $N_\downarrow = 24$ (1/5 configuration), the previous distribution of spinflips persists and the additional 12 spinflips can be arranged on the sites between the hexagons given by 3-site clusters involving two apex spins (for a 3D impression, cf. the blue bonds in Fig. 3). Their total spin is nearly equal to 1/2 and features strong antiferromagnetic correlations (see Fig. 8). This is another stable configuration that resists a change due to the external field.

We note that whenever a localization domain consists out of three sites, as is the case for the sawtooth chain [61–63] or for the tetrahedral chain [71–73], localized magnons naturally form doublets as well. The difference to our case is that the doublets are approximate, appear at a lower field and coexist with the singlets on the squares.

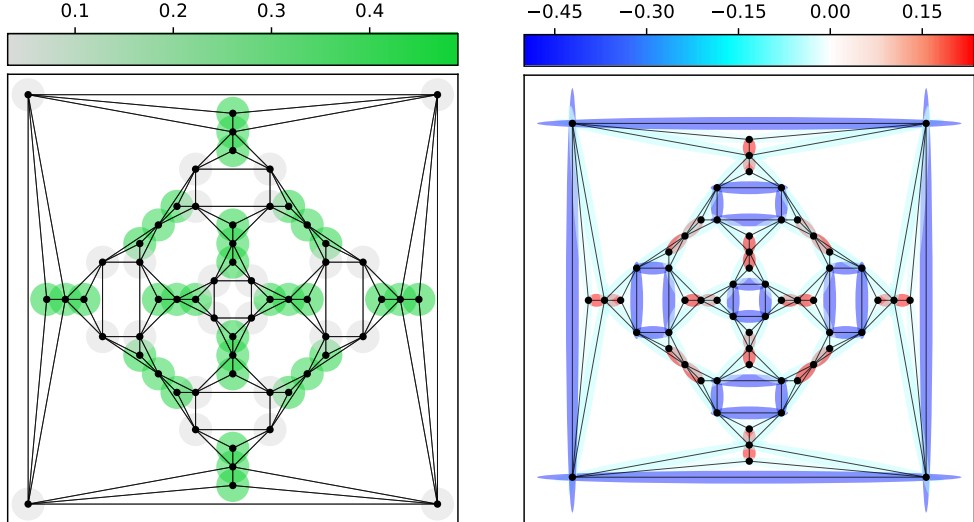

Figure 7: Ground-state properties in the sector $S_{\text{tot}} = 18$ that corresponds to the 3/5 magnetization plateau. The left and right panel show $\langle \mathbf{S}_i \rangle$ and the nearest-neighbour spin-spin correlations, respectively. Note the appearance of localized singlet states, $\langle \mathbf{S}_i \rangle \approx 0$, with strong antiferromagnetic correlations (the grey sites along the square faces in the left picture).

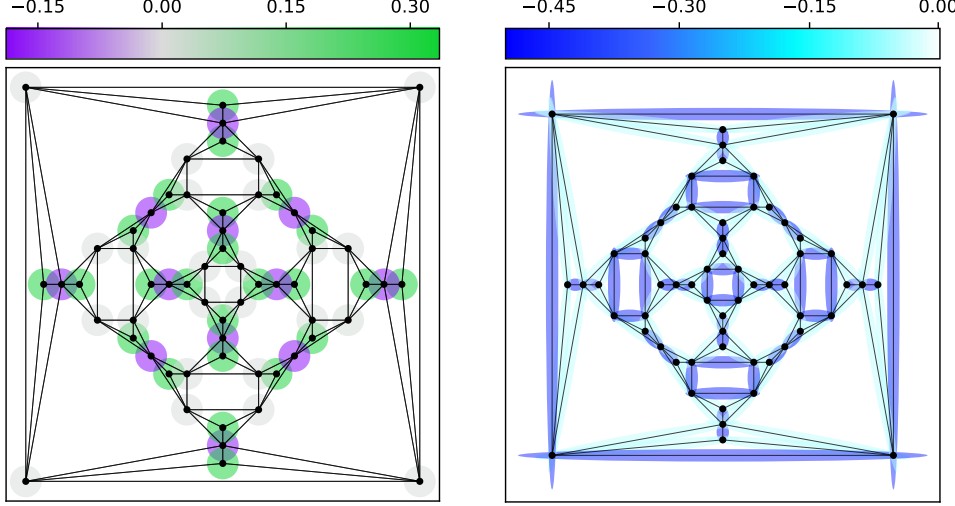

Figure 8: The same as in Fig. 7 but in the sector $S_{\text{tot}} = 6$ that corresponds to the 1/5 magnetization plateau. Note the additional reduction of the local spin on the 12 sites between the hexagon faces (purple). Correspondingly, the 3-site clusters between the squares now acquire a total spin of 1/2 and strong antiferromagnetic correlations.

Overall, we find that the wavefunction at the special fractions of the saturation is again characterized by the notion of disconnection. The 4/5 plateau is governed by 6 independent, localized magnons, which one can show analytically and which is in line with other frustrated geometries. At the 3/5 plateau, the localized-magnon states become 4-site localized singlet states. Finally, at the 1/5 plateau, there is additional room for 12 localized spin-1/2 states.

# 7 Conclusion

We have analyzed the ground-state properties of the antiferromagnetic $S = 1/2$ Heisenberg model on the sodalite cage geometry with 24 vertex-sharing tetrahedra using DMRG. Unlike smaller polyhedra, the ground state is given by the irreducible representation $T$ and is thus threefold degenerate. One can choose each member of the ground-state manifold such that it is symmetry-broken and is invariant only under rotations about one of the three coordinate axes.

The spin-spin correlations signal that the molecule breaks up into three large, nearly disconnected parts (16+16+28 sites). This scenario might be regarded as an extended VBS state, though the disconnection is not exact. Note that an extended-VBS phase with a 12-site unit cell has been recently found on the kagome lattice with second- and third-nearest-neighbour ferromagnetic interactions [38].

The resulting ground states are difficult to connect by local updates with a local Hamiltonian. This entails glass-like behaviour within the DMRG algorithm; standard techniques (such as adding fluctuations) fail, and we need to apply a global operation by explicitly rotating the state.

The physics in the presence of a finite magnetic field is also characterized by confined clusters which lead to magnetization plateaus at special fractions of the saturation. We find localized magnons close to the saturation (4/5) that change into nearly-localized 4-site singlets at the 3/5 plateau. At the 1/5 plateau, they are joined by localized 3-site doublets. These magnetization plateaus are very wide in units of the exchange coupling $J$ and should thus be observable in the experiment.

The results obtained here raise the question whether the ground state for the full 3D pyrochlore lattice may also be crystallized in real space, i.e., breaks the translational symmetry in some nontrivial way, possibly with a large unit cell. As discussed in the introduction, results that show four sublattices have been obtained in the past [24, 25, 27], but this is dissimilar from the SOD60 molecule. Still, we may suspect that systems with vertex-sharing tetrahedra have a general tendency towards spatial symmetry breaking, which manifests itself differently for different geometries. For the SOD60 molecule, in particular, this may be further facilitated by the apex spins, which have a reduced coordination number.

Apart from the connection to the pyrochlores, the results obtained here outline what can be expected from a spin system on the sodalite cage geometry in the extreme quantum limit with $S = 1/2$, in particular regarding potential future experiments.

## Acknowledgements

R.R. and C.K. acknowledge support by the Deutsche Forschungsgemeinschaft (DFG, German Research Foundation) through the Emmy Noether program (KA3360/2-1) as well as by 'Niedersächsisches Vorab' through the 'Quantum- and Nano-Metrology (QUANOMET)' initiative within the project P-1.

M.P. is funded by the Deutsche Forschungsgemeinschaft (DFG, German Research Foundation) – project ID 497779765.

C.P. is supported by the Deutsche Forschungsgemeinschaft (DFG) through the Cluster of Excellence Advanced Imaging of Matter – EXC 2056 – project ID 390715994.

# A  Symmetry transformations for the SOD60 molecule

In order to apply certain symmetry transformations, one has to construct an operator that permutes the sites of the molecule. We are interested in 90° rotations about the three 4-fold symmetry axes connecting the centres of opposite squares. (Alternatively, one could also attempt 120° rotations about the 3-fold symmetry axes, which we did not do.) With respect to the Schlegel projection (see Fig. 1), we define a horizontal (h) axis connecting the left and right square, a vertical (v) axis connecting the lower and upper square, and a perpendicular (p) axis connecting the innermost and outermost square. The corresponding permutations of the index set $\{0,\ldots,59\}$ are listed in Tab. 4. All three permutations decompose into 15 independent cycles, each consisting of three transpositions.

Table 4:   Permutations for the site indices that represent 90° rotations about the specified axes.

| h | v | p |
|---|---|---|
| 0 → 17 | 0 → 30 | 0 → 12 |
| 1 → 31 | 1 → 29 | 1 → 10 |
| 2 → 16 | 2 → 44 | 2 → 24 |
| 3 → 25 | 3 → 15 | 3 → 4 |
| 4 → 32 | 4 → 18 | 4 → 11 |
| 5 → 6 | 5 → 45 | 5 → 25 |
| 6 → 18 | 6 → 46 | 6 → 26 |
| 7 → 24 | 7 → 13 | 7 → 1 |
| 8 → 12 | 8 → 14 | 8 → 0 |
| 9 → 26 | 9 → 5 | 9 → 3 |
| 10 → 47 | 10 → 16 | 10 → 22 |
| 11 → 40 | 11 → 6 | 11 → 9 |
| 12 → 48 | 12 → 17 | 12 → 23 |
| 13 → 2 | 13 → 53 | 13 → 31 |
| 14 → 0 | 14 → 43 | 14 → 17 |
| 15 → 5 | 15 → 54 | 15 → 32 |
| 16 → 29 | 16 → 55 | 16 → 39 |
| 17 → 30 | 17 → 48 | 17 → 41 |
| 18 → 15 | 18 → 56 | 18 → 40 |
| 19 → 10 | 19 → 27 | 19 → 2 |
| 20 → 4 | 20 → 28 | 20 → 6 |
| 21 → 11 | 21 → 20 | 21 → 5 |
| 22 → 39 | 22 → 2 | 22 → 7 |
| 23 → 41 | 23 → 0 | 23 → 8 |
| 24 → 55 | 24 → 31 | 24 → 36 |
| 25 → 46 | 25 → 32 | 25 → 38 |
| 26 → 56 | 26 → 25 | 26 → 37 |
| 27 → 1 | 27 → 42 | 27 → 16 |
| 28 → 3 | 28 → 35 | 28 → 18 |
| 29 → 13 | 29 → 59 | 29 → 47 |
| 30 → 14 | 30 → 58 | 30 → 48 |
| 31 → 44 | 31 → 47 | 31 → 51 |
| 32 → 45 | 32 → 40 | 32 → 52 |
| 33 → 22 | 33 → 19 | 33 → 13 |
| 34 → 23 | 34 → 8 | 34 → 14 |
| 35 → 9 | 35 → 21 | 35 → 15 |
| 36 → 51 | 36 → 1 | 36 → 19 |
| 37 → 38 | 37 → 3 | 37 → 20 |
| 38 → 52 | 38 → 4 | 38 → 21 |
| 39 → 59 | 39 → 24 | 39 → 49 |
| 40 → 54 | 40 → 26 | 40 → 50 |
| 41 → 58 | 41 → 12 | 41 → 34 |
| 42 → 7 | 42 → 33 | 42 → 29 |
| 43 → 8 | 43 → 34 | 43 → 30 |
| 44 → 27 | 44 → 57 | 44 → 55 |
| 45 → 20 | 45 → 50 | 45 → 46 |
| 46 → 28 | 46 → 52 | 46 → 56 |
| 47 → 53 | 47 → 39 | 47 → 57 |
| 48 → 43 | 48 → 41 | 48 → 58 |
| 49 → 36 | 49 → 7 | 49 → 27 |
| 50 → 37 | 50 → 9 | 50 → 28 |
| 51 → 57 | 51 → 10 | 51 → 33 |
| 52 → 50 | 52 → 11 | 52 → 35 |
| 53 → 19 | 53 → 49 | 53 → 44 |
| 54 → 21 | 54 → 37 | 54 → 45 |
| 55 → 42 | 55 → 51 | 55 → 59 |
| 56 → 35 | 56 → 38 | 56 → 54 |
| 57 → 49 | 57 → 22 | 57 → 42 |
| 58 → 34 | 58 → 23 | 58 → 43 |
| 59 → 33 | 59 → 36 | 59 → 53 |

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
