# Peer review of "Magnetic properties of a capped kagome molecule with 60 quantum spins"

_SciPost Physics, doi:SciPost Phys. 12, 143 (2022)_

## Round 1 · Referee Report · Anonymous (Referee 6) · 2021-11-30

Strengths

1) Very interesting model. It has analogies to 2D kagome physics as well as the physics of similar kagome-like magnetic molecules such as V30. The extra ingredient here is the presence of inequivalent spin sites, which I believe is responsible for the specific dimerisation patterns found numerically.

2) Detailed DMRG results have been presented.

Weaknesses

1) Results on dimerization patterns have not been analysed in sufficient detail that would expose the underlying NNVB or RVB physics. See suggestions below.
2) Claims on symmetry breaking are invalid.
3) Claims of spin liquidity are invalid (title and main text).
4) Analogy to dimerisation pattern reported for the pyrochlore is not accurate, see details below.

Report

———— beginning of report————

The Authors present a numerical DMRG study of the ground state properties (in zero and nonzero magnetic fields) of the AF Heisenberg model on a capped kagome molecule with 60 spins S=1/2 (SOD60 in the following). The main results include:

1) a 3-fold degenerate ground state at zero field, with spin-spin correlations showing a clear dimerization pattern and nearly decoupled parts of the molecule.

2) Three wide magnetization plateaus at M=1/5, 3/5 and 4/5. The latter is identified with the physics of localised magnons, which arise from the corner-sharing triangle topology and the associated destructive interference between magnon hoppings. For the 1/5 and 4/5 plateaus the Authors show evidence for polarised spin pattern on top of a specific dimerisation pattern.

3) An appreciable entropy per site at the saturation field, reflecting a 182-fold ground state degeneracy that includes the fully polarised state and the states with 1-, 2-, …, 6-magnons (down to M=4/5), that are related to localized magnons.

These results are interesting and warrant publication in some form but there are a few important aspects that need to be revised before I can recommend the article:

i) The Authors make a statement about a “symmetry-broken ground state” at zero field, based on the fact that the ground state is three-fold degenerate and belongs to one of the 3-fold irreps of Oh.

This statement is invalid. The degeneracy is not enough for symmetry breaking. First of all, any linear combination of the three ground states is also a ground state. The correlation patterns shown in Fig. 2 are obtained using the combinations that break the symmetry explicitly. One could also calculate the correlation patterns of the symmetric combinations (Fig. 3). The DMRG code (as any variational algorithm that is based on local operations/updates) has a tendency to deliver the combinations that break the symmetry, but this does not mean that the symmetry can be broken in this physical system. For the spontaneous symmetry breaking to occur one would need a macroscopically large system, whereas here we only have 60 spins (in some dynamical sense, one can argue that 60 spins is big enough for a slow rotation of the correlation patterns from one panel of Fig. 2 to another in time, but strictly speaking we need a macroscopically large system for that time to become infinitely large).

ii) the Authors refer to the system as a spin liquid (and, at the same time, they make the conflicting statement that the symmetry is broken, which is also not valid, see point i above). However, there is no indication for spin liquidity here. In particular, the ground state degeneracy is not associated to any topological index, that one would normally associate with a spin liquid. Otherwise, I do not see why we should not call every magnetic molecule a spin liquid. I would therefore suggest that the Authors remove the term spin-liquid from the title.

iii) In their conclusions, the Authors make a comparison between the dimerisation pattern presented here with that seen in previous studies in the S=1/2 AF in the pyrochlore lattice, and suggest that the origin of the dimerization in the two cases is similar. This comparison is not valid: In the present case the dimerisation is enforced by the open boundaries of the system (red spins in Fig. 1. left) and the presence of inequivalent symmetry points more generally, whereas in the pyrochlore system the dimerizations proposed in previous studies arise spontaneously by a symmetry breaking of the space group symmetry.

iv) Strictly speaking, the ground state degeneracy at the saturation field is 182 and not 181, if we are to include the fully polarized state. So the entropy of S=ln(181) needs to be replaced by S=ln(182).

Suggestions for further analysis:
I would also like to propose a couple of suggestions which may improve the presentation and add further important insights:

a) Since the presented results point to a specific dimerization patterns, one would expect that a simple examination of the “nearest neighbour valence bond” coverings of the molecule could reveal some insights for the origin of these patterns. For example, the decoupling between the different parts of the molecule can possibly be explained by checking the percentage of NNVB coverings that cross the dotted boundaries of fig. 2. Can the Authors enumerate the NNVB coverings and calculate e.g. the average correlations and compare with the DMRG results?

b) Related to above:
The outer spins on the boundary of SOD60 (red spins in Fig. 1) have a smaller coordination number compared to inner sites, so one could expect a stronger dimerization on the bonds that involve the outer spins. I believe this can be seen in the presented correlation patterns, and the Authors could comment on this.

c) One could also comment on the impact of tunneling between different NNVB configurations. Naively, one would expect that resonances of singlets around squares are stronger compared to resonances around hexagons. Looking at Fig. 1a, this would imply that the red spins would couple stronger to the green spins (forming strong singlets) rather than to the blue spins, thereby leaving the latter to participate into square resonances to lower the energy further. I believe the correlation pattern of Fig. 3 shows this, and the Authors could comment on it.

d)Regarding the decoupling between large parts of the molecule:
Since the S.S correlations across the dotted boundaries of Fig. 2 are “very small” (how small? the Authors could provide some numbers here), one could reproduce the ground state energy and ground state configuration patterns by doing exact diagonalizations on the separate parts. I believe this could reproduce the ground state energy within the accuracy related to the spin-spin correlations across the dotted boundaries.

———— end of report————

  • validity: good
  • significance: good
  • originality: good
  • clarity: good
  • formatting: good
  • grammar: excellent

Author:  Roman Rausch  on 2022-03-01  [id 2252]

(in reply to Report 2 on 2021-11-30)

We would like to thank the referee for evaluating our work, the constructive criticism and the useful suggestions. We agree with most points, but we would still argue that we have a symmetry-broken state (with some caveats for finite systems, which are of general nature).

The point-by-point reply is as follows:

The referee's comment:

>>>>>>>>>>>>>>>>
i) The Authors make a statement about a “symmetry-broken ground state” at zero field, based on the fact that the ground state is three-fold degenerate and belongs to one of the 3-fold irreps of Oh.

This statement is invalid. The degeneracy is not enough for symmetry breaking. First of all, any linear combination of the three ground states is also a ground state. The correlation patterns shown in Fig. 2 are obtained using the combinations that break the symmetry explicitly. One could also calculate the correlation patterns of the symmetric combinations (Fig. 3). The DMRG code (as any variational algorithm that is based on local operations/updates) has a tendency to deliver the combinations that break the symmetry, but this does not mean that the symmetry can be broken in this physical system. For the spontaneous symmetry breaking to occur one would need a macroscopically large system, whereas here we only have 60 spins (in some dynamical sense, one can argue that 60 spins is big enough for a slow rotation of the correlation patterns from one panel of Fig. 2 to another in time, but strictly speaking we need a macroscopically large system for that time to become infinitely large).
>>>>>>>>>>>>>>>>

Our reply:

The statement that symmetry breaking requires the thermodynamic limit refers to energy barriers at finite temperatures. To find common ground on this question, one can actually go to an even simpler case, such as the Ising ferromagnet, where everything is under control analytically. In this case, any finite system will fluctuate between the configurations with m=1/2 and m=-1/2, with zero magnetization on the average. However, the barrier in the free energy between these two configurations becomes infinite in the thermodynamic limit. So if the system is initially positioned in one of the minima, it gets trapped there forever.

The referee already gives a solution of how to modify these ideas for finite systems: Instead of the fluctuation time being infinite, it just becomes very large for large systems and at low temperatures. With these disclaimers, we maintain that it is still useful to think of our results as showing symmetry breaking. This is especially so at T=0.

While one can argue that T=0 does not strictly exist, neither do infinite systems. Hence, we have to believe that symmetry breaking, which in the theory exactly emerges only in the thermodynamic limit, can in some sense be approximated in reality by ever larger systems.

That any linear combination within the ground-state manifold is also a ground state is of course generally valid. Thus, it is always possible to "homogenize" the correlation functions. For example, in the simple case of the Majumdar-Ghosh chain, one can superimpose the even-odd and odd-even dimerization patterns to get translationally invariant spin-spin correlations. However, to get the correct value for the correlations, one also needs the orthogonal complement to this superposition and has to perform an average over the two results, analogous to we did in Table 2.

DMRG converges to a particular linear combination that is largely random, though a state with small entanglement is preferred. This is irrelevant here, however, since we have already obtained the full ground-state manifold.

Note that Fig. 3 is *not* obtained from a symmetric linear combination of the three ground states, but rather as an average over expectation values (Eq. (6)). This can be written as a trace over an overlap matrix, which ensures that it is independent of the choice of basis within the ground-state manifold, i.e. of the particular linear combination.

We have added a disclaimer to the revised manuscript to be more careful about the issue of symmetry breaking.

The referee's comment:

>>>>>>>>>>>>>>>>
ii) the Authors refer to the system as a spin liquid (and, at the same time, they make the conflicting statement that the symmetry is broken, which is also not valid, see point i above). However, there is no indication for spin liquidity here. In particular, the ground state degeneracy is not associated to any topological index, that one would normally associate with a spin liquid. Otherwise, I do not see why we should not call every magnetic molecule a spin liquid. I would therefore suggest that the Authors remove the term spin-liquid from the title.
>>>>>>>>>>>>>>>>

Our reply:

This is largely a question of language and definitions. The definition of a QSL is for example discussed in the review by Savary & Balents (2017) and we do not see a that a definitive conclusion is drawn. In their view "not all QSLs are topological, and not all topological phases are QSLs". A relation to finite systems is not discused. An older working definition by Chayes, Chayes & Kivelson (1989) is "disordered state with both translational and spin-rotational symmetries", which is applicable to finite systems.

As we have explained in the introduction, the ground states of frustrated polyhedra would be "liquid" in this latter sense. Clearly, not every magnetic molecule fulfils that - there are ferromagnetic and ferrimagnetic ones, and those with dimerization or quasi-long-range order.

To avoid confusion, we have now changed the title of the paper to "Magnetic properties of a capped kagome molecule with 60 quantum spins", but we note that this is somewhat inadequate as well, since the ground state is actually non-magnetic. We have also examined all uses of "spin liquid" in the paper and replaced them by more careful formulations.

The referee's comment:

>>>>>>>>>>>>>>>>
iii) In their conclusions, the Authors make a comparison between the dimerisation pattern presented here with that seen in previous studies in the S=1/2 AF in the pyrochlore lattice, and suggest that the origin of the dimerization in the two cases is similar. This comparison is not valid: In the present case the dimerisation is enforced by the open boundaries of the system (red spins in Fig. 1. left) and the presence of inequivalent symmetry points more generally, whereas in the pyrochlore system the dimerizations proposed in previous studies arise spontaneously by a symmetry breaking of the space group symmetry.
>>>>>>>>>>>>>>>>

Our reply:

There seems to be some confusion here: While there are three inequivalent types of lattice sites, this is *not* the symmetry breaking we are talking about. The effect is rather that geometrically equivalent bonds show different correlations.
We point out that no symmetry breaking occurs for the smaller SOD20 molecule (Fig. 4), despite having similar inequivalent sites. This means that, for SOD20, geometrically equivalent bonds show the same correlations.

We agree insofar as the inequivalent lattice sites may to some degree *facilitate* spatial symmetry breaking, but we would not say that they "enforce" it.

The comparison with the pyrochlore lattice clearly suggests itself because of the geometry. Of course, a lattice cannot just split into three parts like the molecule, but rather into sublattices, possibly with large unit cells. We thought that our formulation was already quite careful, but we have now revised it to be even more careful.

The referee's comment:

>>>>>>>>>>>>>>>>
iv) Strictly speaking, the ground state degeneracy at the saturation field is 182 and not 181, if we are to include the fully polarized state. So the entropy of S=ln(181) needs to be replaced by S=ln(182).
>>>>>>>>>>>>>>>>

Our reply:

We thank the referee for spotting that mistake. We have corrected it in the revised version.

The referee's suggestion:

>>>>>>>>>>>>>>>>
a) Since the presented results point to a specific dimerization patterns, one would expect that a simple examination of the “nearest neighbour valence bond” coverings of the molecule could reveal some insights for the origin of these patterns. For example, the decoupling between the different parts of the molecule can possibly be explained by checking the percentage of NNVB coverings that cross the dotted boundaries of fig. 2. Can the Authors enumerate the NNVB coverings and calculate e.g. the average correlations and compare with the DMRG results?
>>>>>>>>>>>>>>>>

Our reply:

This is an interesting idea. We have evaluated the Hafnian of the J-matrix, which corresponds to the number of NNVB states and obtain 5,971,817 for SOD60. For the 16-site poles, we get 2 and for the 28-site belt we get 800. This means that there should be just 2*2*800=3600 NNVB configurations that do not cross the boundaries, or 0.06%. So the vast majority of the NNVB basis states actually do cross the boundaries, which does not seem to explain the spatial pattern.

However, the NNVB basis may not be helpful here, as it is not linearly independent for systems of tetrahedra (cf. Wildeboer & Seidel, 2011). Linear independence is already violated locally for a single tetrahedron, which one can check analytically.

Addressing the question of whether the NNVB may somehow still work well for tetrahedra-based systems is clearly out of the scope of this work. We have added a discussion of this to the revised manuscript under subsection 5.1.

The referee's suggestion:

>>>>>>>>>>>>>>>>
b) Related to above:
The outer spins on the boundary of SOD60 (red spins in Fig. 1) have a smaller coordination number compared to inner sites, so one could expect a stronger dimerization on the bonds that involve the outer spins. I believe this can be seen in the presented correlation patterns, and the Authors could comment on this.
>>>>>>>>>>>>>>>>

Our reply:

We are not entirely sure what is meant here by "stronger dimerization", but in all the VBS-like patterns in Fig. 2, only the weak red-blue bonds (apex-square) are cut, while the strong red-green bonds (apex-hexagon) remain.

This is related to the next comment:

>>>>>>>>>>>>>>>>
c) One could also comment on the impact of tunneling between different NNVB configurations. Naively, one would expect that resonances of singlets around squares are stronger compared to resonances around hexagons. Looking at Fig. 1a, this would imply that the red spins would couple stronger to the green spins (forming strong singlets) rather than to the blue spins, thereby leaving the latter to participate into square resonances to lower the energy further. I believe the correlation pattern of Fig. 3 shows this, and the Authors could comment on it.
>>>>>>>>>>>>>>>>

Our reply:

Indeed, as seen from Tab. 2, the largest correlations are found between red and green sites in the average values and they are never cut in the disconnected patterns. Blue-blue bonds (on the squares) come second in terms of strength. On the other hand, the red-blue (apex-square) correlations are not very strong.
We agree that from an RVB perspective, one would expect that parallel bonds resonate, which would explain the relatively high correlations of the blue-blue bonds at the expense of red-blue ones.
We have added a discussion of this into the revised manuscript (subsection 5.1).

The referee's suggestion:

>>>>>>>>>>>>>>>>
d)Regarding the decoupling between large parts of the molecule:
Since the S.S correlations across the dotted boundaries of Fig. 2 are “very small” (how small? the Authors could provide some numbers here), one could reproduce the ground state energy and ground state configuration patterns by doing exact diagonalizations on the separate parts. I believe this could reproduce the ground state energy within the accuracy related to the spin-spin correlations across the dotted boundaries.
>>>>>>>>>>>>>>>>

Our reply:

For the particular ground states, The red-blue correlations are roughly -0.027 and the blue-green ones are roughly -0.0076. They contribute about 0.018*J to the energy density (or around 4%).
Computing the energy for the separate parts is an interesting idea that is easy to implement. With this procedure we obtain [2*E_0(pole)+E_0(equator)]/L=-0.4294 or about 99.5% of the exact ground-state energy density.
We note that our approach (at the beginning of Sec. 5) is in terms of the total spin of the separated parts, which also includes longer-distance spin-spin correlations not shown in the paper. But estimating the energy in the way suggested byt the referee is also an interesting idea, which we have included into the revised manuscript.

---

## Round 1 · Referee Report · Anonymous (Referee 4) · 2021-12-14

Strengths

interesting new magnetic model
ground state established convincingly
innovative idea of constructing degenerate ground states
additional study of the magnetization curve reveals localized magnons
DMRG state-of-the-art and beautiful figures

Weaknesses

no exact experimental realization of model known today

Report

The authors study the spin-1/2 Heisenberg model on a three-dimensional frustrated geometry using the density matrix renormalization group. The physics of the Heisenberg model on this particular geometry has not been previously studied. Even though a direct experimental system appears not to be currently known, the model geometry is motivated well by the authors.

The performed DMRG simulations are currently state-of-the-art and allow the authors to reach a convincing answer to the question about the nature of the ground state. The observed symmetry breaking is an interesting finding. I particularly liked their idea to construct the three ground states related via a rotation by implementing swap operators of the full rotation. It is interesting how closely degenerate these three states are, as can be read in Table 1, given the fact that the simulated system "only" has 60 lattice sites. I think the term "symmetry-breaking" is therefore adequate, even though we are on a finite system.

The authors call this state a "generalization" of a valence bond solid. However, it is not fully clear in what sense this constitutes a generalization. It would be beneficial if the authors could elaborate on this. Moreover, the authors might consider adding Phys. Rev. B 102, 020411(R) (2020) to their list of references, where a VBS has been established on a similar geometry.

Following the discussion of the ground state, the physics at finite magnetic field is studied. Also here, the authors find convincing answers to the occurrence of magnetization plateaux at specific values of the saturation magnetization. The degeneracies of the localized magnon states are worked out in detail and compared to other relevant models.

Overall, I find that the study investigates an interesting model for which clear answers to its ground state physics are given by state-of-the-art DMRG simulations. As such, I can recommend publication in SciPost.

I detected one small type, the authors might want to correct. In line 190, Fig. 1 is referenced, whereas I think this should rather be Table 1.

  • validity: high
  • significance: high
  • originality: high
  • clarity: top
  • formatting: perfect
  • grammar: excellent

Author:  Roman Rausch  on 2022-03-01  [id 2254]

(in reply to Report 4 on 2021-12-14)

We would like to thank the referee for the positive evaluation of our work and for the recommendation for publication. The comments are adressed as following:

The referee's comment:

>>>>>>>>>>>>>>>>
The authors call this state a "generalization" of a valence bond solid. However, it is not fully clear in what sense this constitutes a generalization. It would be beneficial if the authors could elaborate on this. Moreover, the authors might consider adding Phys. Rev. B 102, 020411(R) (2020) to their list of references, where a VBS has been established on a similar geometry.
>>>>>>>>>>>>>>>>

Our reply:

The typical VBS state, as found in the Majumdar-Ghosh chain, has only pair-singlets or dimers. What we mean by "generalization" is just an extension to larger clusters of tetramers, hexamers and so on. (In our case, the regions encompass 16+16+28 sites.)
A better term is probably "extended", which is also used in the paper cited by the referee. The paper is indeed relevant to our case, showing VBS states extended over 12 sites.
We have revised this in the new manuscript and added the reference.

The referee's comment:

>>>>>>>>>>>>>>>>
I detected one small type, the authors might want to correct. In line 190, Fig. 1 is referenced, whereas I think this should rather be Table 1.
>>>>>>>>>>>>>>>>

Our reply:

Indeed. This has been corrected in the revised manuscript.

---

## Round 1 · Referee Report · Anonymous (Referee 5) · 2021-12-14

Strengths

1- Novel quantum spin cluster with interesting magnetization curve.
2- Rigorous analysis of a few ground states within the concept of localized magnons.
3- Manuscript is concise and clearly written.

Weaknesses

1- The usage of the term “spin liquid” is inappropriate for relatively small finite-size quantum spin cluster and this term should be treated with much greater caution.
2- Model without direct experimental realization.

Report

Authors studied ground-state properties of a capped kagome molecule with 60 magnetic centers. They have used density matrix renormalization group for the diagonalization of the Hamiltonian and they present field-dependence of the magnetization at zero temperature. I also appreciate the explanation of ground states through the concept of localized magnons.
In my opinion, the results presented in this manuscript are scientifically sound, novel and interesting. Hence, I consider the present paper appropriate for SciPost Physics journal, because the presented results may be beneficial from theoretical point of view. Nevertheless, I have a few remarks and suggestions, which should be considered by the authors before the manuscript will be accepted for publication.

Requested changes

(1) The main emphasis of the present paper is laid on spin-liquid properties of the capped kagome molecule. To the best of my knowledge, the spin-liquid state is not possible for any finite-size system. According to this fact, the authors should be more careful when using this term and I recommend to reformulate few statements including the term “spin-liquid” in the title, abstract and main text (e. g. page 3, line 84-87).

(2) It is not exactly clear if threefold degeneracy and broken symmetry are present just because of the used DMRG method, or one can indeed expect them in the studied system. Authors should clarify this issue in somewhat more detail.

(3) I do not understand, why authors used the term “Magic values” for the fractional values of the widest plateaus in the magnetization curve. If specifically, 4/5-plateau is nearly of the same width as 2/3-plateau following the widest 3/5-plateau. Authors should clarify this poetic term if they want to keep it.

(4) Page 12, line 297-299 and 313-314. The authors argue that within 3/5 plateau there appears singlet state localized on six square faces (plaquettes ), which can be alternatively viewed nearly as localized two-magnon state. Please know that this feature has been recently reported for several quantum Heisenberg octahedral chains when extending the concept of localized magnons in order to cover full magnetization curve from zero up to saturation field (e.g. Strecka and al, PRB 95, 224415, 2017). The aforementioned paper and related other papers reporting localized two-magnon states should be mentioned in this context.

(5) Page 12, line 305-307 It would be valuable if the authors could comment in more detail character of highly nontrivial localized doublets spread over 3-site clusters. Are these localized doublets quite analogous to the case of sawtooth chain when localized magnon is located in the valley?

(6) For the benefit for the reader it will be advisable to add on page 4 new figure for SOD20 molecule, which will be quite analogous to Fig 1.

(7) For better clarity, Fig. 3 should have the same coloring of sites as Fig. 1.

(8) First row in Tab 3: number of magnons should be zero.

(9) Could the authors comment in the manuscript why there are just 13 linearly independent localized one-magnon state and not 14 (i.e the total number of squares and hexagons)?

(10) Since the DMRG technique is appropriate mainly for 1D quantum spin systems, it would be advisable to check numerical precision of this method when comparing at least of a few particular lowest-energy eigenstates where full exact diagonalization or Lanczos diagonalization could be applied (e.g. for the sector with S_T=18 corresponding to sizeable 3/5 plateau.)

(11) A few minor points of formal character only:
-Abstract is too long with respect with requirement of the journal.
-place the abbreviation VBS on line 10 before the word “state”.
-“stark contrast” on line 10 should be replaced with “sharp contrast”
-A few hyphens are missing, e.g., ground-state energy on line 57, localized-magnon states on line 313, etc.
-Page 2, Line 61 replace the word “see” with “indicate”
-Page 6, line 180 the word “following” should be replaced with word “permutation”
-in the conclusion authors should used on line 318-319 the abbreviation DMRG instead of full description.

  • validity: high
  • significance: good
  • originality: good
  • clarity: high
  • formatting: good
  • grammar: excellent

Author:  Roman Rausch  on 2022-03-01  [id 2253]

(in reply to Report 3 on 2021-12-14)

We would like to thank the referee for reading our paper and finding our work "sound, novel and interesting". Here is our point-by-point reply to the comments and suggestions.

The referee's comment:

>>>>>>>>>>>>>>>>
(1) The main emphasis of the present paper is laid on spin-liquid properties of the capped kagome molecule. To the best of my knowledge, the spin-liquid state is not possible for any finite-size system. According to this fact, the authors should be more careful when using this term and I recommend to reformulate few statements including the term “spin-liquid” in the title, abstract and main text (e. g. page 3, line 84-87).
>>>>>>>>>>>>>>>>

Our reply:

This depends on the definition of "spin liquid", of course. But since it appears that our definition is in disagreement with two referees, we have replaced all uses of "spin liquid" in the paper by more careful formulations.

The referee's comment:

>>>>>>>>>>>>>>>>
(2) It is not exactly clear if threefold degeneracy and broken symmetry are present just because of the used DMRG method, or one can indeed expect them in the studied system. Authors should clarify this issue in somewhat more detail.
>>>>>>>>>>>>>>>>

Our reply:

The degeneracy is not an artifact of the numerical method, but is actually what is found for the solution of the model. DMRG is a controlled method that is closer to exact diagonalization (albeit with some truncation error for finite bond dimension) rather than to approximative methods that make systematic neglections.

The main smoking-gun evidence for the degeneracy is the lack of respect of the spatial symmetries for a single ground state. For that to happen, we see no other way than (a) the accuracy is too low or (b) the ground state is degenerate and one has converged to some particular superposition of basis states from the ground-state manifold that does not have to respect the symmetry.
Case (a) can be excluded by ensuring a small energy variance (or a large enough bond dimension), as we did. One can gather further evidence for case (b) by constructing the full multiplet, as we also did.

We have added a similar explanation to the revised manuscript.

The referee's comment:

>>>>>>>>>>>>>>>>
(3) I do not understand, why authors used the term “Magic values” for the fractional values of the widest plateaus in the magnetization curve. If specifically, 4/5-plateau is nearly of the same width as 2/3-plateau following the widest 3/5-plateau. Authors should clarify this poetic term if they want to keep it.
>>>>>>>>>>>>>>>>

Our reply:

"Magic" is used here in analogy to the "magic numbers" of the electron shells. The meaning of "magic" here is just "special" and maybe "not obvious" or "not easily predicatable".
But since the use of this word seems to be confusing in this context, we have omitted it in the revised manuscript.

The referee's comment:

>>>>>>>>>>>>>>>>
(4) Page 12, line 297-299 and 313-314. The authors argue that within 3/5 plateau there appears singlet state localized on six square faces (plaquettes ), which can be alternatively viewed nearly as localized two-magnon state. Please know that this feature has been recently reported for several quantum Heisenberg octahedral chains when extending the concept of localized magnons in order to cover full magnetization curve from zero up to saturation field (e.g. Strecka and al, PRB 95, 224415, 2017). The aforementioned paper and related other papers reporting localized two-magnon states should be mentioned in this context.
>>>>>>>>>>>>>>>>

Our reply:

We have included these papers into our citations and made an appropriate comment in the revised manuscript.

The referee's comment:

>>>>>>>>>>>>>>>>
(5) Page 12, line 305-307 It would be valuable if the authors could comment in more detail character of highly nontrivial localized doublets spread over 3-site clusters. Are these localized doublets quite analogous to the case of sawtooth chain when localized magnon is located in the valley?
>>>>>>>>>>>>>>>>

Our reply:

Whenever the localization domain for magnons consists out of three sites, flipping a spin and placing a magnon naturally creates a localized doublet. So what we have in the molecule looks very similar to what is found in the sawtooth chain and the tetrahedral chain.
However, in our case, the doublets appear at a lower field, are only approximate and coexist with the singlet states on the squares.

We have added a comment about this with the appropriate citations into the revised manuscript.

The referee's comment:

>>>>>>>>>>>>>>>>
(6) For the benefit for the reader it will be advisable to add on page 4 new figure for SOD20 molecule, which will be quite analogous to Fig 1.
>>>>>>>>>>>>>>>>

Our reply:

We have added such a figure in the revised manuscript.

The referee's comment:

>>>>>>>>>>>>>>>>
(7) For better clarity, Fig. 3 should have the same coloring of sites as Fig. 1.
>>>>>>>>>>>>>>>>

Our Reply:

Unfortunately, the red/green/blue colours for the sites clash with the colormap of the bonds in this case, which in our mind reduces the clarity. We have tried to alter the colormap, but were not able to come to a result that does not look confusing.

As a compromise, we now repeat a part of the molecule with the red/green/blue balls on the right side of Fig. 3 for easier reference.

The referee's comment:

>>>>>>>>>>>>>>>>
(8) First row in Tab 3: number of magnons should be zero.
>>>>>>>>>>>>>>>>

Our Reply:

This is amended in the revised manuscript.

The referee's comment:

>>>>>>>>>>>>>>>>
(9) Could the authors comment in the manuscript why there are just 13 linearly independent localized one-magnon state and not 14 (i.e the total number of squares and hexagons)?
>>>>>>>>>>>>>>>>

Our Reply:

We have added such an explanation to the revised manuscript, as well as a sketch to illustrate it in Fig. 5.
The short answer: The reason is that the squares and hexagons share sites with each other. A staggered sum over the hexagon-magnons then becomes proportional to a sum of square-magnons, meaning that one of the 14 states is linearly dependent.

The referee's comment:

>>>>>>>>>>>>>>>>
(10) Since the DMRG technique is appropriate mainly for 1D quantum spin systems, it would be advisable to check numerical precision of this method when comparing at least of a few particular lowest-energy eigenstates where full exact diagonalization or Lanczos diagonalization could be applied (e.g. for the sector with S_T=18 corresponding to sizeable 3/5 plateau.)
>>>>>>>>>>>>>>>>

Our Reply:

This subspace is too hefty for our exact diagonalization code that only exploits the U(1) symmetry and does not exploit the point-symmetry groups (Hilbert space size: 1.4*10^12).
However, we have validated the code by comparing the ground-state energy of the icosidodecahedron (L=30, S=1/2, Hilbert space size: 1.6*10^8), which is known using ED (Rousochatzakis et al., 2008):
ED: -0.44114054
Chi(SU2)=500: -0.44101247, variance/site=6.5*10^-4
Chi(SU2)=1000: -0.44112922, variance/site=7.3*10^-5
Chi(SU2)=2000: -0.44113998, variance/site=4.4*10^-6
The variance/site for SOD60 was around 10^-4 to 10^-5. Thus we expect an accuracy within 3-4 digits past the decimal point, which is consistent with Table 1 from the paper.

Furthermore, we have checked that DMRG reproduces the ground-state degeneracies for the localized magnon sectors down to Stot=24 using Eq. (4) from the paper (though exact diagonalization is faster and more accurate in this case). We have added a corresponding comment to the revised manuscript.

The referee's comment:

>>>>>>>>>>>>>>>>
(11) A few minor points of formal character only:
-Abstract is too long with respect with requirement of the journal.
-place the abbreviation VBS on line 10 before the word “state”.
-“stark contrast” on line 10 should be replaced with “sharp contrast”
-A few hyphens are missing, e.g., ground-state energy on line 57, localized-magnon states on line 313, etc.
-Page 2, Line 61 replace the word “see” with “indicate”
-Page 6, line 180 the word “following” should be replaced with word “permutation”
-in the conclusion authors should used on line 318-319 the abbreviation DMRG instead of full description.
>>>>>>>>>>>>>>>>

We thank the referee for spotting these. We have shortened the abstract to 200 words in the revised manuscript and made the suggested corrections.

---

## Round 2 · Referee Report · Anonymous (Referee 3) · 2022-3-3

Report

The authors have properly addressed the critiques raised by the referees. Their revised manuscript is an excellent paper and I can fully recommend publication in SciPost.

---

## Round 2 · Referee Report · Anonymous (Referee 2) · 2022-3-9

Report

The authors have taken into consideration all suggestions and remarks from my previous reviewer report. Accordingly, the manuscript has been substantially improved and I therefore recommend the present version of the manuscript for publication.

---

## Round 2 · Referee Report · Anonymous (Referee 1) · 2022-3-14

Strengths

see previous report

Weaknesses

see previous report

Report

—— beginning of report ——

I have read the revised draft and the point by point reply of the Authors. The Authors have removed the conflicting term “spin liquid” and have added a discussion on what they mean by “symmetry broken” state for this finite-size system. They have also incorporated other suggestions which have improved the content of the paper.

I would therefore like to recommend the paper for publication.

Further comments/suggestions:

1) On the notion of “symmetry broken” state, the Authors may wish to further clarify sentences such as:

line 8: “We find a threefold degenerate ground state that breaks the spatial symmetry” lines 95-96: “… making it threefold degenerate and thus in principle symmetry-broken.”

Such statements give the general impression that degeneracy implies symmetry breaking. Perhaps it would help if the Authors clarify that what they mean is that the ground state manifold contains different irreps (angular momentum L=0, 2pi/3 and -2pi/3) of the C3 subgroup, associated with one of the four C3 axes of the molecule. As long as these have different irreps, a linear combination can break the C3 symmetry (in the finite-size sense).

2) The Authors may wish to add a few sentences about the symmetry group and perhaps illustrate one of the three-fold axes in Fig. 1, as this is the symmetry that is broken (again, in the finite-size sense).

3) line 133: “...has the irreducible representations A (1), E (2), T (3)”: The group Oh has 10 irreps in total: A1g(1), A1u(1), A2g(1), A2u(1), Eg(2),Eu(2),T1g(3) ,T1u(3),T2g(3),T2u(3). Perhaps the Auhors meant "classes of irreps"?

4) line 166: “...three rotational symmetry axes that pierce the square faces”: The three-fold axes pierce the hexagon faces (and therefor connect one square face to another) and not the square faces. I guess this is a typo.

5) regarding the NNVB picture and the intuitive explanation of the different strengths of NN spin-spin correlations: Perhaps the Authors can still add a few sentences and explicitly mention that the spin-spin correlations on squares are much stronger than those on hexagons due to stronger tunneling, but still weaker than some of the outer bonds, which have strong dimerisation due to the apical dangling spins.

— end of report —

Requested changes

Further comments/suggestions:

1) On the notion of “symmetry broken” state, the Authors may wish to further clarify sentences such as:

line 8: “We find a threefold degenerate ground state that breaks the spatial symmetry” lines 95-96: “… making it threefold degenerate and thus in principle symmetry-broken.”

Such statements give the general impression that degeneracy implies symmetry breaking. Perhaps it would help if the Authors clarify that what they mean is that the ground state manifold contains different irreps (angular momentum L=0, 2pi/3 and -2pi/3) of the C3 subgroup, associated with one of the four C3 axes of the molecule. As long as these have different irreps, a linear combination can break the C3 symmetry (in the finite-size sense).

2) The Authors may wish to add a few sentences about the symmetry group and perhaps illustrate one of the three-fold axes in Fig. 1, as this is the symmetry that is broken (again, in the finite-size sense).

3) line 133: “...has the irreducible representations A (1), E (2), T (3)”: The group Oh has 10 irreps in total: A1g(1), A1u(1), A2g(1), A2u(1), Eg(2),Eu(2),T1g(3) ,T1u(3),T2g(3),T2u(3). Perhaps the Auhors meant "classes of irreps"?

4) line 166: “...three rotational symmetry axes that pierce the square faces”: The three-fold axes pierce the hexagon faces (and therefor connect one square face to another) and not the square faces. I guess this is a typo.

5) regarding the NNVB picture and the intuitive explanation of the different strengths of NN spin-spin correlations: Perhaps the Authors can still add a few sentences and explicitly mention that the spin-spin correlations on squares are much stronger than those on hexagons due to stronger tunneling, but still weaker than some of the outer bonds, which have strong dimerisation due to the apical dangling spins.

  • validity: good
  • significance: good
  • originality: good
  • clarity: good
  • formatting: good
  • grammar: excellent

Author:  Roman Rausch  on 2022-05-06  [id 2447]

(in reply to Report 3 on 2022-03-14)

>On the notion of “symmetry broken” state, the Authors may wish to further clarify sentences such as:
>
>line 8: “We find a threefold degenerate ground state that breaks the spatial symmetry”
>lines 95-96: “… making it threefold degenerate and thus in principle symmetry-broken.”
>
>Such statements give the general impression that degeneracy implies symmetry breaking. Perhaps it would help if the Authors clarify that what they mean is that the ground state manifold contains different irreps (angular momentum L=0, 2pi/3 and -2pi/3) of the C3 subgroup, associated with one of the four C3 axes of the molecule. As long as these have different irreps, a linear combination can break the C3 symmetry (in the finite-size sense).

We would say that symmetry breaking implies degeneracy, but not vice versa (topological degeneracy is a prime counterexample), so we agree that the statement is somewhat misleading and we have revised it in the final version.
However, the ground-state manifold does not contain several irreps. Rather, we have a nontrivial, 3-dimensional irrep.

>2) The Authors may wish to add a few sentences about the symmetry group and perhaps illustrate one of the three-fold axes in Fig. 1, as this is the symmetry that is broken (again, in the finite-size sense).

We have illustrated the C4 and C3 symmetry axes in Fig. 1 of the final manuscript.

>3) line 133: “...has the irreducible representations A (1), E (2), T (3)”:
>The group Oh has 10 irreps in total: A1g(1), A1u(1), A2g(1), A2u(1), Eg(2),Eu(2),T1g(3) ,T1u(3),T2g(3),T2u(3).
>Perhaps the Auhors meant "classes of irreps"?

This has been improved in the final version.

>4) line 166: “...three rotational symmetry axes that pierce the square faces”:
>The three-fold axes pierce the hexagon faces (and therefor connect one square face to another) and not the square faces. I guess this is a typo.

This is not a typo: There are both 3-fold axes and 4-fold axes. The 4-fold axes are easier to recognize on the square projection, so we were mainly thinking and writing in terms of them.
To be precise, the symmetry is broken such that two of the three 4-fold axes become 2-fold axes, while the 3-fold axes cease to be symmetry axes altogether. We have included this explanation into the final manuscript.

>5) regarding the NNVB picture and the intuitive explanation of the different strengths of NN spin-spin correlations:
>Perhaps the Authors can still add a few sentences and explicitly mention that the spin-spin correlations on squares are much stronger than those on hexagons due to stronger tunneling, but still weaker than some of the outer bonds, which have strong dimerisation due to the apical dangling spins.

We believe that our statement expresses exactly this:
"In the case of SOD60, parallel bonds are found on the square plaquettes (blue-blue) and this may explain their relatively large correlations [...] at the expense of the red-blue and green-blue ones. This leaves the red (apex) spins to couple more strongly with the green spins."
Therefore, we have chosen not to do further changes in this case.

We would like to once more thank the referee for a thorough reading of our work and for these final comments.

---

## Round 2 · List of Changes

• change of title to "Magnetic properties of a capped kagome molecule with 60 quantum spins"
  • shortening of the abstract to 200 words
  • incorporation of various new references as suggested by the referees
  • correction of S=ln(181) to S=ln(182)
  • change of all usages of "spin liquid"
  • "magic fraction" removed
  • remark that the degeneracy is not a numerical artifact
  • remark on symmetry breaking in finite systems
  • sketch of the neighbourhood of a tetrahedron in Fig. 3
  • more ways to quantitatively characterize the disconnection patterns in Sec. 5
  • ball-and-stick drawing of the SOD20 molecule in Fig. 4
  • new subsection 5.1: "Nearest-neighbour valence bond picture"
  • sketch of the hexagon localization domains in Fig. 5
  • discussion of why there are 13 instead of 14 localized magnons in Sec. 6.1
  • discussion of the relationship of the localized doublets and singlets to findings from other papers
  • conclusion: added Ref. [38] and changed the discussion of the comparison to the pyrochlore lattice
  • addition of clarifying remarks regarding the geometry and the computational details
  • a number of various small chages: typos, hyphens, word replacements etc. (often as pointed out by the referees)

---

## Editorial Decision

published